# Effect of Multiantibiotic-Loaded Bone Cement on the Treatment of Periprosthetic Joint Infections of Hip and Knee Arthroplasties—A Single-Center Retrospective Study

**DOI:** 10.3390/antibiotics13060524

**Published:** 2024-06-03

**Authors:** Benedikt Paul Blersch, Florian Hubert Sax, Moritz Mederake, Sebastian Benda, Philipp Schuster, Bernd Fink

**Affiliations:** 1Department of Joint Replacement, General and Rheumatic Orthopaedics, Orthopaedic Clinic Markgröningen gGmbH, Kurt-Lindemann-Weg 10, 71706 Markgröningen, Germany; benedikt.blersch@rkh-gesundheit.de (B.P.B.); florian.sax@rkh-gesundheit.de (F.H.S.); philipp.schuster@rkh-gesundheit.de (P.S.); 2Department of Trauma and Reconstructive Surgery, BG Klinik, University of Tübingen, Schnarrenbergstraße 95, 72076 Tübingen, Germany; mmederake@bgu-tuebingen.de; 3Department of Trauma, Hand Surgery and Orthopedics, Clinic Konstanz, Mainaustraße 35, 78464 Konstanz, Germany; sebastian.benda@glkn.de; 4Department of Orthopaedics and Traumatology, Paracelsus Medical University, Prof. Ernst Nathan Straße 1, 90419 Nuremberg, Germany; 5Orthopaedic Department, University Hospital Hamburg-Eppendorf, Martinistrasse 52, 20246 Hamburg, Germany

**Keywords:** septic revision arthroplasty, two-stage revision, antibiotic-impregnated cement, admixed antibiotic, easy to treat, difficult to treat

## Abstract

Background: Two-stage septic revision is the prevailing method for addressing late periprosthetic infections. Using at least dual-antibiotic-impregnated bone cement leads to synergistic effects with a more efficient elution of individual antibiotics. Recent data on the success rates of multiantibiotic cement spacers in two-stage revisions are rare. Methods: We conducted a retrospective follow-up single-center study involving 250 patients with late periprosthetic hip infections and 95 patients with prosthetic knee infections who underwent septic two-stage prosthesis revision surgery between 2017 and 2021. In accordance with the antibiotic susceptibility profile of the microorganisms, a specific mixture of antibiotics within the cement spacer was used, complemented by systemic antibiotic treatment. All patients underwent preoperative assessments and subsequent evaluations at 3, 6, 9, 12, 18, and 24 months post operation and at the most recent follow-up. Results: During the observation period, the survival rate after two-step septic revision was 90.7%. Although survival rates tended to be slightly lower for difficult-to-treat (DTT) microorganism, there was no difference between the pathogen groups (easy-to-treat (ETT) pathogens, methicillin-resistant staphylococci (MRS), and difficult-to-treat (DTT) pathogens). Furthermore, there were no differences between monomicrobial and polymicrobial infections. No difference in the survival rate was observed between patients with dual-antibiotic-loaded bone cement without an additional admixture (Copal^®^ G+C and Copal^®^ G+V) and patients with an additional admixture of antibiotics to proprietary cement. Conclusion: Employing multiple antibiotics within spacer cement, tailored to pathogen susceptibility, appears to provide reproducibly favorable success rates, even in instances of infections with DTT pathogens and polymicrobial infections.

## 1. Introduction

Periprosthetic joint infections (PJIs) represent significant complications following total hip and knee arthroplasty, occurring at a rate of 1–2% [1,2,3]. The genesis of periprosthetic infections is multifactorial: risk factors for periprosthetic infections include an increased body mass index (BMI) [4,5,6,7], increased age [8], female gender [9], diabetes mellitus [5,6,8,9], rheumatoid arthritis [6,9], previous joint surgery [6,8], previous PJI [7,8], hypothyroidism [9], preoperative high-dose steroids [5,6], chronic alcohol use [7,9], tobacco use [5,7], coronary artery disease [5], HIV infection at an advanced stage [8], depression [6], and the presence of distant infectious foci [8]. In cases of late-onset PJIs (occurring more than 4 weeks post operation), the complete removal of all foreign materials in a septic one- or two-stage revision procedure is necessary.

The two-stage septic revision was described by Insall et al. in 1983 [10], and the success rates have improved in recent decades: despite success rates below 90% in the 1980s (65% in Rand et al. for two-stage septic knee revision [11], and 87% in McDonald et al. for two-stage septic hip revision [12]), more current studies indicate success rates well above 90% [13,14,15].

The two-stage revision protocol involves an initial intervention to remove foreign material and to perform radical debridement, followed mostly by the implantation of a spacer containing antibiotics. This is followed by an interim period of typically 6 to 12 weeks before the explantation of the spacer, the reimplantation of a cemented or uncemented prosthesis, and the subsequent administration of antibiotics for a further 6 to 12 weeks [1]. Antibiotics, tailored to pathogen sensitivity, are incorporated into the cement of the spacer [1]. Two-stage septic revision remains the most employed method for addressing late periprosthetic infections.

Multidrug-resistant bacteria complicate adequate antibiotic therapy. In particular, the antibiotic treatment of PJIs caused by biofilm-forming pathogens that are resistant to biofilm-active antibiotics is a particular challenge: because fluoroquinolones target biofilm-forming Gram-negative bacteria and rifampicin targets biofilm-forming Gram-positive bacteria, rifampicin-resistant pathogens and fluoroquinolone-resistant Gram-negative bacteria are considered difficult-to-treat bacteria [16,17].

The use of dual antibiotics in the bone cement of a spacer produces a synergistic effect, resulting in the improved elution of individual components compared to the use of a single antibiotic [18,19,20]. By determining antibiotic concentrations in the membrane around hip spacers with Copal G+C cement and admixed vancomycin in Copal cement, Fink et al. [21] were able to show that, even 6 weeks after implantation, the release of all three antibiotics was higher than the minimum inhibitory concentration for the pathogens causing a PJI; therefore, in our clinic, we regularly use bone cement with two antibiotics (Copal G+C or Copal G+V (Heraeus Medical, Werheim, Germany)) for septic revision surgery and often add a third antibiotic depending on the pathogen and its sensitivity. Little amounts of data have been published to date on the success rates of these multiantibiotic cement spacers in two-stage revisions.

Therefore, the aim of this study was to answer the following questions:What is the success rate of two-stage revision with the dual-antibiotic-loaded bone cement Copal G+C and Copal G+V?What is the success rate of the two-stage revision with the dual-antibiotic-loaded bone cement Copal G+C and Copal G+V with a further admixed antibiotic?Are there differences in the success rate for the different groups of pathogens: those that are easy-to-treat (ETT), methicillin-resistant (MRS), and difficult-to-treat (DTT)?Are there factors that influence the success rate of two-stage septic revision surgery?

## 2. Results

Thirty patients (9.3%) showed reinfection during the course of treatment and underwent another septic revision. This resulted in a survival rate of the prothesis of 90.7% during the observation period of 47.2 ± 20.9 (24.0–112.0) months (Figure 1).

There was no difference between patients with two antibiotics in the cement without an additional admixture (Copal^®^ G+C and Copal^®^ G+V) and those with an additional admixture of antibiotics to proprietary cement (Copal^®^ G+C, Copal^®^ G+V, and Palacos^®^ +G with an additional cement admixture) (Table 1). There was no difference between the pathogen groups, but the survival rate tended to be slightly worse for the difficult-to-treat (DTT) pathogens (Table 1, Figure 2). There was also no difference between monomicrobial and polymicrobial infections (Table 1). No significantly worse survival curve was seen in patients with a previous spacer change, but there was in patients who had previously undergone aseptic or septic revision surgery (Table 1).

Otherwise, there were no differences between hip and knee prosthesis replacements with respect to gender and whether or not diabetes mellitus or another immune-modulating underlying disease was present (Table 1).

In 19 of the 30 patients with reinfection (63.3%) a pathogen shift was observed. In cases of a pathogen shift, an increase in pathogenicity was noted in eight patients (26.7%): a shift from an ETT pathogen to a DTT pathogen occurred in five cases (16.7%) and a shift from an ETT pathogen to an MRS pathogen was observed in two cases (6.7%). A decrease in pathogenicity, manifested as a shift from a DTT pathogen to an MRS pathogen, a DTT pathogen to an ETT pathogen, or an MRS pathogen to an ETT pathogen, was observed in only one case each (Table 2).

Identical pathogens were detected in five patients with reinfection (16.7%); there was no case with a change in pathogenicity (Table 3).

In the remaining six cases of reinfection (20.0%), either a culture-negative infection was initially identified and in reinfection a pathogen was detected (four cases, 13.3%), or after initial pathogen detection a culture-negative reinfection occurred (two cases, 6.7%; Table 4).

## 3. Discussion

Overall, the concept with multiple-antibiotic admixtures in the spacer cement showed a very good success rate of 90.7%. Interestingly, we achieved good success rates for all pathogen groups (ETT, MRS, and DTT), with only a slightly lower success rate for difficult-to-treat (DTT) pathogens. The outcomes in the present study correspond to those of Faschingbauer et al. [16]. Faschingbauer et al. [16] saw no statistically significant difference in the recurrent infection rates between organism groups of difficult-to-treat (DTT) (19.5%), methicillin-resistant staphylococci (MRS) (16.7%), and easy-to-treat (ETT) bacteria (15.4%) in 137 two-stage revisions of infected knee arthroplasties, using a static spacer with clindamycin in 26.3% and vancomycin in 18.9%. In contrast, Kurd et al. [22] found higher reinfection rates for periprosthetic infections with methicillin-resistant staphylococci, and Citak et al. [23] as well as Rossmann et al. [24] found an infection with enterococci to be a risk factor for reinfection for one-stage septic knee revisions. Abdelaziz et al. [25] found the same for one-stage septic hip TEP revisions. We were unable to identify a higher risk of reinfection for MRSA pathogens or enterococci in our study, nor did we see that polymicrobial periprosthetic infections have worse success rates than periprosthetic infections with only one pathogen, as has been shown in several previous studies [26,27].

The use of multiple antibiotics in the spacer cement, which are specifically tailored to the pathogens and often include an additional local antibiotic admixture to the cement as well, may play a role in our outcomes. We saw a significantly lower overall reinfection rate than Faschingbauer et al. [16], with 8.9% for ETT, 9.1% for MRS, and 12.9% for DTT pathogens in our study. Faschingbauer et al. [16] had used only proprietary cement with one antibiotic in 54.8% and two antibiotics in 45.2% of cases. Our outcomes are in line with those of Chalmers et al. [28], who saw a success rate of 92% at 2 years and 88% at 5 years in 135 two-stage hip TEP revisions with cementless reimplantation using a custom-made spacer similar to ours, and, routinely, two antibiotics in the cement with vancomycin and gentamycin.

The beneficial impact of cement containing a combination of two or more antibiotics, such as gentamicin (G) and clindamycin (C), in the treatment of PJIs can be attributed to several key factors:According to the spectrum of activity, the combination of at least two antibiotics demonstrates synergistic effects, leading to an extended range of effectiveness. This explains that virtually all pathogens associated with periprosthetic infections (PJIs) are targeted by a minimum of two antibiotics [20,29,30].The elution of individual antibiotics is more efficient when they are present together in the ALBC compared to when they are used individually within the cement. The molecules of all antibiotics used are small, hydrophilic, and present excellent diffusion properties [30].As a result of the elevated local antibiotic release, the typically bacteriostatic antibiotic clindamycin attains local concentrations that manifest bactericidal activity [30].The effect mechanism or target of the antibiotics gentamicin and clindamycin in bacteria is synergistic: gentamicin targets the 30S ribosome on the mRNA and clindamycin the 50S ribosome. This simultaneous attack on bacteria at two different sites results in synergy that prevents the development of bacterial resistance [30].

Ensing et al. demonstrated in vitro that dual-antibiotic-impregnated cement, Copal G+C, exhibited greater efficacy in inhibiting bacterial biofilm formation compared to the single-impregnated cement Palacos R+G [19]. Frew et al. [31] were able to show that adding vancomycin to cement containing gentamycin resulted in a fivefold higher release of vancomycin and a two-fold higher release of gentamycin than the two antibiotics in proprietary Copal G+V cement.

Like Faschingbauer et al. [16], we did not classify periprosthetic joint infection with MRSA and MRSE as difficult-to-treat, because studies using levofloxacin and rifampicin in combination showed success rates of up to 91% in experimental and 93.1% in one-stage septic revisions [32,33,34]. Kurd et al. [22] present opposite results in studying the rate of failure in two-stage revision knee arthroplasty: failed two-stage revision arthroplasty was 3.37 times more likely in PJIs originally caused by methicillin-resistant organisms. The patients were routinely treated with a static spacer containing a vancomycin and tobramycin admixture in Palaco’s R+G cement, but only 3 out of 30 cases were treated systemically with rifampicin. Kramer et al. [35] found a significantly lower reinfection rate in staphylococcal-induced periprosthetic infections when rifampicin was administered during septic revisions.

Yang et al. [36] reported a success rate of 63.4% in 41 two-stage revisions of fungal periprosthetic joint infections of the knee with amphotericin B in the spacer cement. Kim et al. [37] found a significant correlation between fungal infections and an increased reinfection rate in two-stage septic knee prosthesis replacements. We only had two cases with fungal infection that were treated successfully.

Kubista et al. [38] and Faschingbauer et al. [16] identified a previous revision between the first operation and the final reimplantation as a risk factor for reinfection in two-stage septic knee prosthesis replacements. We did not observe this in the present study, but the number of exchanges of the spacer were relatively low; however, we saw significantly worse survival curves in the present study if aseptic or septic revision surgery had occurred previously. Previous revision surgery has been shown to be a risk factor for reinfection in several studies [39,40]. Citak et al. [23] as well as Rossmann et al. [24] found previous septic revisions to be a risk factor for the failure of one-stage septic knee prosthesis replacements.

Reinfection of PJIs can be caused by the same microorganism or by a different germ. The first scenario was seen in five cases in our study and can be explained by an insufficient treatment of the first PJI. A reason for unsuccessful treatment can be the ability of microorganisms to survive intracellular inside human cells. These microorganisms can be the source of an exacerbation of a PJI, especially in conditions of a depressed immune status of a human [41]. Persistent osteitis can be another explanation for these reinfections [42,43].

The second scenario, with a reinfection of a PJI with another microorganism, was seen in our study in 19 cases. This can firstly be explained by the fact that the first PJI was induced by several germs, where one was the leading microorganism that could be treated successfully, but not the second. The second could survive and was the cause of the reinfection. Secondly, it can be a superinfection, for example, with a difficult-to-treat pathogen that was not treated with antibiotics and is responsible for the reinfection. This is described by Darwich et al. 2021 [17]. We observed a shift from an ETT to a DTT pathogen in five cases (16.7%), and a shift from an ETT to an MRS pathogen in three cases (10.0%), which could be partly explained by superinfections. Thirdly, it could be a real new infection after successful treatment of the first PJI. Patients with comorbidities and a high risk for PJIs would also have a higher risk for a new infection. All cases of a shift of pathogens of the current study could be theoretically explained by a real new infection.

This study has some weaknesses: Despite the larger number of two-stage septic revision surgeries, the number of patients with DTT and MRS pathogens was low, at 31 and 33 patients, respectively. This may also be a reason for the lack of significance between the three pathogen groups. Moreover, the number of infections with specific difficult-to-treat microorganisms (Enterococcae, fungal infection) were too low for statistical analysis. Therefore, the hypothesis and outcomes of this study should be verified by multicenter studies with even higher numbers of patients.

In summary, the use of multiple antibiotics in spacer cement, tailored to the susceptibility of the pathogens, appears to provide reproducibly favorable success rates even in polymicrobial infections and infections with difficult-to-treat pathogens. This observation still needs to be confirmed in further studies with larger numbers of cases (preferably multicenter studies).

Alongside adding antibiotics to the bone cement, there are other newer concepts with which to prevent biofilm formation on implants. For example, modern silver coatings of implants have the potential to reduce biofilm formation [44]. Future studies have to show whether this coating alone or in combination with antibiotic-loaded cement is more effective. Moreover, cementless titanium implants can be coated with antimicrobial hydrogels to reduce biofilm formation [45]; however, this approach also requires validation through future clinical studies.

## 4. Materials and Methods

The study is a retrospective single-center study conducted at the Orthopaedic Clinic Markgröningen (Germany).

In it, 250 patients (72.5%) with late periprosthetic infection of the hip endoprosthesis and 95 (27.5%) patients with PJIs of the knee arthroplasty underwent septic two-stage prosthesis revision surgery between 2017 and 2021. The patient cohort consisted of 158 females (45.8%) and 187 males (54.2%), aged 69.4 ± 10.8 (27.8–96.8) years. The average body mass index (BMI) was 29.8 ± 6.5 kg/m^2^ (17.7–60.6). The time between primary surgery to revision with prosthesis explantation and cement spacer implantation was 91.7 ± 87.7 (2–500) months. With respect to the American Society of Anesthesiologists (ASA) classification scores, 12 patients (3.5%) were classed as ASA 1, 155 patients (44.9%) were ASA 2, 169 patients (49.0%) were ASA 3, and 9 patients (2.6%) were ASA 4 [46,47]. Regarding the Charlson Comorbidity Index (CCI), there were 78 patients (22.6%) with CCI 0, 49 patients (14.2%) with CCI 1, 46 patients (13.3%) with CCI 2, 47 patients with CCI 3 (13.6%), 50 patients (14.5%) with CCI 4, 36 patients (10.4%) with CCI 5, 24 patients (7.0%) with CCI 6, 8 patients (2.3%) with CCI 7, 3 patients (0.9%) with CCI 8, 2 patients (0.6%) with CCI 9, 1 patient (0.3%) with CCI 10, and 1 patient (0.3%) with CCI 11 [46,48]. With regard to secondary diseases potentially relevant to the development of PJIs, 65 patients (18.8%) had diabetes mellitus and 14 patients (4.1%) had an inflammatory disease: 5 patients (1.4%) were suffering from rheumatic arthritis, 3 patients (0.9%) had spondylitis ankylosans, 3 patients (0.9%) had Morbus Crohn, and 1 patient (0.3%) each had colitis ulcerosa, polymyalgia rheumatica, and psoriasis. At least one previous septic operation was carried out in 83 patients (24.1%), and at least one previous aseptic operation was carried out in 78 patients (22.6%). In 25 patients (7.2%) an exchange of the spacer was performed, because the systemic infection parameters (CRP > 40 mg/L) were suspected to not calm the infection.

The diagnosis of periprosthetic infection followed the guidelines established by the Musculoskeletal Infection Society (MSIS) and ICM 2018 [49,50]. Preoperative aspiration, representing a standard practice at our clinic, was conducted and the bacteriological culture of the aspirated fluid was monitored for 14 days in accordance with Schäfer et al. [51]. In cases of unclear diagnosis, a biopsy of the periprosthetic tissue was performed, where five tissue samples were taken for bacteriological and five tissue samples for histological examination, in addition to another joint aspiration. Bacteriological and histological analyses, following the protocols outlined in Atkins et al. [52], Virolainen et al. [53], and Pandey et al. [54], were conducted on the membrane extracted from the loosened site during the surgical procedure to verify the initial diagnosis. The microorganisms identified using these procedures are detailed in Table 5; it is remarkable that two responsible organisms were detected in 56 instances, and three in 6 instances (polymicrobial infection in 62 cases, 18.0%). The microorganisms were divided into three groups according to Faschingbauer et al. [16]: group 1 are difficult-to-treat (DTT) organisms, including quinolone-resistant Gram-negative bacteria, rifampicin-resistant staphylococcus, enterococcus, and candida; group 2 are methicillin-resistant staphylococci (MRS), including methicillin-resistant *staphylococcus areus* (MRSA) and methicillin-resistant *staphylococcus epidermidis* (MRSE); and group 3 are easy-to-treat (ETT) bacteria, including all other microorganism- and culture-negative periprosthetic joint infections (PJIs).

According to the antibiotic susceptibility profile of the microorganisms, a specific mixture of antibiotics was recommended by our microbiologist for use in the cement of the spacer (Table 6) and in the systemic treatment (Table 7 and Table 8).

The cement of the spacer contained two antibiotics in 84 cases (24.3%) and three antibiotics in 261 cases (75.7%): proprietary Copal G+C [Heraeus Medical, Wehrheim, Germany] alone was used in 69 patients (20.0%), and proprietary Copal G+V [Heraeus Medical, Wehrheim, Germany] in 9 patients (2.6%). Additional antibiotics up to a maximum of 10% *w*/*w* of the total cement powder were added to the proprietary Palacos G [Heraeus Medical, Wehrheim, Germany] in 6 cases (1.7%), to Copal G+C cement [Heraeus Medical, Wehrheim, Germany] in 254 cases (73.6%), and to Copal G+V cement [Heraeus Medical, Wehrheim, Germany] in 7 cases (2.0%).

The parenteral antibiotic treatment was designed by our microbiologist for each case and initiated intraoperatively subsequent to the extraction of the implant, effective debridement of infected and ischemic tissues, and the harvesting of a minimum of five tissue specimens for bacteriological evaluation (enrichment for 14 days). The tissue samples were obtained from the joint capsule, the membrane surrounding the loosened region, and the suspected infected tissues.

The intravenously administered antibiotics are listed in Table 7; there were 5 cases (1.4%) where one antibiotic was administered systemically, 325 cases (94.2%) with two antibiotics, 13 cases (3.8%) with three antibiotics, and 2 cases (0.6%) with four antibiotics.

An individual custom-made interim hip prosthesis was implanted following the previous descriptions [55,56]. The stem spacer unit consisted of a cemented prosthesis stem encased with an antibiotic-supplemented cement and coated with the patient’s autologous blood in order to support an easier removal. The cemented spacer was implanted using cement mixed six minutes prior to the implantation to reduce the quality of cement interdigitation and to simplify the subsequent removal of the spacer–cement in the second-stage operation. The femoral and acetabular components of the spacer were articulated with a metal head.

The acetabular spacer consisted of a polyethylene cup that was cemented into either a Ganz ring (229 cases, 91.6%) or a Burch–Schneider acetabular reinforcement ring (21 cases, 8.4%) (ZimmerBiomet, Winterthur, Switzerland). The Ganz ring and the Burch–Schneider acetabular reinforcement ring were fixed with two to four screws and a customized antibiotic mixture in the cement tailored to the specific microbial susceptibility. The two-stage septic femoral revision was performed using either a transfemoral approach in 149 cases (59.6%) or an endofemoral approach in 101 cases (40.4%). The transfemoral and the endofemoral reimplantation was conducted using cementless femoral stems. Hereby, at the explantation of the spacer and reimplantation, the transfemoral approach was reopened, the stem was reimplantated, and the approach was closed using new double cerclage wires with a diameter of 1.5 mm.

Articulating spacers for the knee were preformed using molds (Heraeus Medical, Werheim, Germany) in 51 cases and static spacers were constructed using rods of an external fixator in 44 cases. In all cases a rotating hinge was cemented in at the second stage.

While performing the reimplantation procedure, a minimum of five tissue samples were taken for bacterial examination and the initial parenteral antibiotic treatment was resumed. The parenteral antibiotic therapy was maintained for two weeks before switching to the appropriate oral antibiotic administration for a further four-week period. As an exception, rifampicin and ciprofloxacin were administered orally on the second day post surgery, as was the case previously.

The orally administered antibiotics are listed in Table 8; one antibiotic was administered in 9 cases (2.6%), two antibiotics were administered in 329 cases (95.4%), three antibiotics were administered in 6 cases (1.7%), and four antibiotics were administered in 1 case (0.3%).

All patients underwent preoperative examinations, as well as postoperative assessments at 3 months, 6 months, 9 months, 1 year, 18 months, and 2 years following the surgery, and at the most recent follow-up. Inflammatory markers, especially C-reactive protein levels, were monitored. Based on the criteria established by Masri et al. [57] and Zimmerli et al. [58], patients were considered free from infection during the follow-up period if they showed no clinical signs of infection (elevated temperature, local pain, redness, warmth, and sinus tract infection) and obtained a C-reactive protein level below 10 mg/L.

Retrospective data collection and analysis were performed. Statistical analysis was conducted using IBM SPSS Statistics for Windows (version 24, IBM Corp., Armonk, NY, USA). The Kaplan–Meier method was employed to calculate the cumulative survival, and the resulting survival curve was illustrated as a Kaplan–Meier plot. The survival analysis was conducted using the log-rank test. A significance level of alpha < 0.05 was considered significant. Data presentation included mean ± standard deviation (and range) or number (percentage), unless specified otherwise. Informed consent was obtained from all participants, and the study protocol was approved by the research ethics boards of the Landesärztekammer Nord-Württemberg (F-2023-115).

## Figures and Tables

**Figure 1 antibiotics-13-00524-f001:**
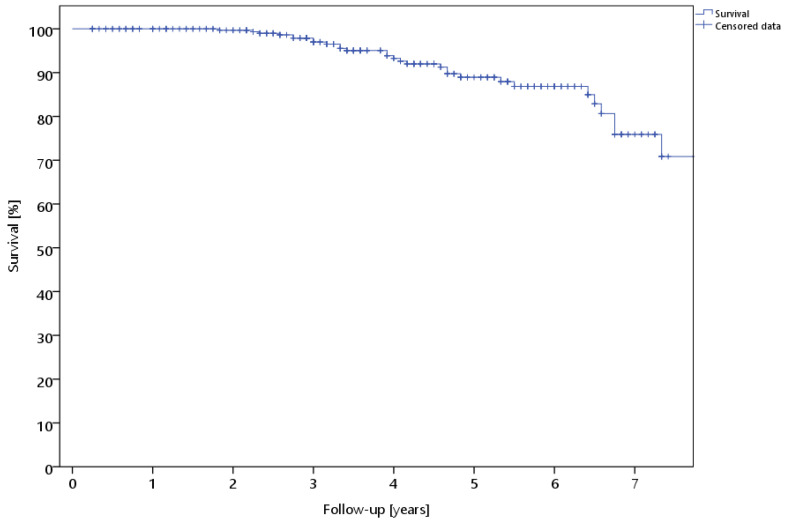
Survival rate of the prothesis of all patients with septic re-revision as the endpoint.

**Figure 2 antibiotics-13-00524-f002:**
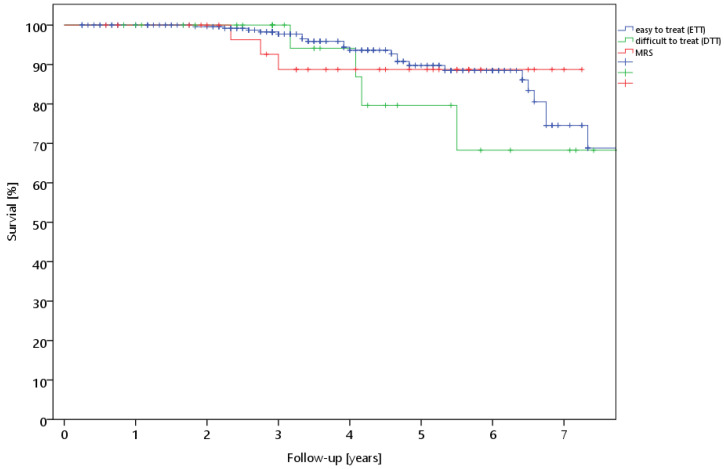
Survival rate of the prothesis for the different groups of bacteria (ETT = easy-to-treat, DTT = difficult-to-treat, and MRS = methicillin-resistant staphlococci).

**Table 1 antibiotics-13-00524-t001:** Five-year survival (Kaplan–Meier) and log-rank test (* statistically significant).

Factor	Five-Year Survival (Kaplan–Meier)	Log-Rank Test
Joint		0.743
Hip	87.4 [80.9–93.9]	
Knee	91.7 [85.2–98.2]	
Sex		0.103
Male	82.0 [74.2–89.8]	
Female	96.8 [93.3–100.0]	
Bacteria		0.646
Easy-to-treat (ETT)	89.9 [84.7–94.9]	
Difficult-to-treat (DTT)	79.6 [58.8–100.0]	
MRS	88.7 [76.7–100.0]	
Diabetes Mellitus		0.468
Yes	88.3 [78.3–98.3]	
No	89.1 [83.8–94.4]	
Monobacterial infection	87.8 [82.5–93.1]	0.550
Polybacterial infection	94.5 [86.9–100.0]	
Septic Reoperation After Primary Implantation		0.012 *
Yes	75.6 [63.3–87.6]	
No	93.5 [89.4–97.6]	
Aseptic Reoperation After Primary Implantation		0.001 *
Yes	71.2 [46.7–95.7]	
No	91.3 [86.8–95.8]	
Changing of Spacer		0.350
Yes	81.5 [58.2–100.0]	
No	89.5 [84.8–94.2]	
Used Cement		0.268
Copal^®^ G+C or Copal^®^ G+V	94.6 [88.3–100.0]	
Copal^®^ G+C or Copal^®^ G+V or	87.1 [81.2–93.0]	
Palacos^®^ +G with additional		
antibiotics mixed by hand		

**Table 2 antibiotics-13-00524-t002:** Cases of reinfection with a pathogen shift.

	Microorganism One (First Septic Revision)	Microorganism Two (First Septic Revision)	Microorganism One (Second Septic Revision)	Microorganism Two (Second Septic Revision)
1	*Streptococcus agalactiae* (ETT)		*Staphylococcus aureus* (ETT)	
2	*Staphylococcus aureus* (ETT)		*Enterococcus faecalis* (DTT)	*Citrobacter koseri* (ETT)
3	*Streptococcus mitis* (ETT)	*Streptococcus oralis* (ETT)	*Staphylococcus epidermidis* (MRS)	
4	*Staphylococcus epidermidis*, rifampicin-resistant *staphylococcus* (DTT)		*Enterobacter cloacae* (ETT)	
5	*Pseudomonas aeruginosa* (ETT)		*Staphylococcus epidermidis* (MRS)	*Candida albicans* (DTT)
6	*Streptococcus gordonii* (ETT)		*Streptococcus mutans* (ETT)	
7	*Streptococcus mitis* (ETT)		*Staphylococcus epidermidis* (MRS)	
8	*Staphylococcus capitis* (ETT)	*Staphylococcus epidermidis* (ETT)	*Staphylococcus lugdunensis* (ETT)	
9	*Staphylococcus aureus* (ETT)		*Raouletta planticola* (ETT)	
10	*Staphylococcus caprae* (ETT)		*Staphylococcus epidermidis* (ETT)	
11	*Staphylococcus saprophyticus* (ETT)		*Staphylococcus epidermidis*, rifampicin-resistant *staphylococcus* (DTT)	
12	*Staphylococcus aureus* (ETT)		*Staphylococcus epidermidis*, rifampicin-resistant *staphylococcus* (DTT)	
13	*Staphylococcus capitis* (ETT)		*Corynebacterium urealyticum* (ETT)	*Cutibacterium acnes* (ETT)
14	*Cutibacterium acnes* (ETT)		*Cutibacterium granulosum* (ETT)	
15	*Staphylococcus capitis* (ETT)	*Cutibacterium acnes* (ETT)	*Staphylococcus epidermidis*, rifampicin-resistant *staphylococcus* (DTT)	
16	*Staphylococcus epidermidis* (MRS)		*Cutibacterium acnes* (ETT)	
17	*Cutibacterium acnes* (ETT)		*Staphylococcus capitis* (ETT)	
18	*Cutibacterium acnes* (ETT)		*Staphylococcus aureus* (ETT)	
19	*Candida albicans* (DTT)		*Staphylococcus epidermidis* (MRS)	

**Table 3 antibiotics-13-00524-t003:** Cases of reinfection with identical pathogens.

	Microorganism One (First Septic Revision)	Microorganism Two (First Septic Revision)	Microorganism (Second Septic Revision)
1	*Staphylococcus aureus* (ETT)	*Finegoldia magna* (ETT)	*Staphylococcus aureus* (ETT)
2	*Staphylococcus aureus* (ETT)		*Staphylococcus aureus* (ETT)
3	*Staphylococcus aureus* (ETT)		*Staphylococcus aureus* (ETT)
4	*Staphylococcus epidermidis* (MRS)		*Staphylococcus epidermidis* (MRS)
5	*Eschericia coli*, quinolone-resistant Gram-negative bacteria (DTT)		*Eschericia coli*, quinolone-resistant Gram-negative bacteria (DTT)

**Table 4 antibiotics-13-00524-t004:** Cases of reinfection with culture-negative infection in first or second septic revision.

	Microorganism One (First Septic Revision)	Microorganism One (Second Septic Revision)
1	No pathogen detectable	*Pseudomonas aeruginosa* (ETT)
2	No pathogen detectable	*Staphylococcus aureus* (ETT)
3	No pathogen detectable	*Staphylococcus aureus* (MRS)
4	No pathogen detectable	*Streptococcus dysgalacticae* (ETT)
5	*Cutibacterium acnes* (ETT)	No pathogen detectable
6	*Enterococcus faecalis* (DTT)	No pathogen detectable

**Table 5 antibiotics-13-00524-t005:** Identified microorganisms and the number of detections.

	Microorganism	Number	% of Cases Infected by This Pathogen
Easy-to-treat (ETT) bacteria	*Staphylococcus epidermidis*	67	19.42
	*Staphylococcus aureus*	51	14.78
	*Propionibacterium acnes* (*Cutibacterium acnes*)	43	12.46
	*Staphylococcus capitis*	17	4.93
	*Propionibacterium granulosum*	12	3.48
	*Staphylocccus hominis*	12	3.48
	*Staphylococcus lugdunensis*	9	2.61
	*Staphylococcus caprae*	7	2.03
	*Streptococcus mitis*	7	2.03
	*Eschericia coli*	5	1.45
	*Staphylococcus haemolyticus*	5	1.45
	*Staphylococcus warneri*	5	1.45
	*Streptococcus agalactiae*	5	1.45
	*Streptococcus anginosus*	4	1.16
	*Streptococcus oralis*	4	1.16
	*Listeria monocytogenes*	3	0.87
	*Staphylococcus saprophyticus*	3	0.87
	*Streptococcus gordonii*	3	0.87
	*Streptococcus salivarius*	3	0.87
	*Actinomyces neuii ssp.*	2	0.58
	*Corynebacterium jekeium*	2	0.58
	*Corynebacterium species*	2	0.58
	*Corynebacterium striatum*	2	0.58
	*Klebsiella pneumoniae*	2	0.58
	*Peptostreprococcus micros*	2	0.58
	*Pseudomonas aeruginosa*	2	0.58
	*Actinomyces odontolyticus*	1	0.29
	*Bacteroides fragilis*	1	0.29
	*Citrobacter koseri*	1	0.29
	*Corynebacterium amycolatum*	1	0.29
	*Corynebacterium minutissimum*	1	0.29
	*Dermabacter hominis*	1	0.29
	*Enterobacter aerogenes*	1	0.29
	*Enterobacter sakazakii*	1	0.29
	*Finegoldia magna*	1	0.29
	*Granulicatella adicems*	1	0.29
	*Lactobacillum plantarum*	1	0.29
	*Mycobacterium tuberculosis*	1	0.29
	*Peptostreptococcus magnus*	1	0.29
	*Propionibacterium propionicum*	1	0.29
	*Proteus mirabilis*	1	0.29
	*Rhizobium radiobacter*	1	0.29
	*Staphylococcus chromogenes*	1	0.29
	*Staphylococcus simultans*	1	0.29
	*Streptoccocus sanguinis*	1	0.29
	*Streptococcus dysgalacticae*	1	0.29
	*Streptococcus parasanguis*	1	0.29
Methicillin-resistant staphylococci (MRS)	*Staphylococcus epidermidis* (MRSE)	31	8.99
	*Staphylocccus hominis* (MRS)	1	0.29
	*Staphylococcus capitis* (MRS)	1	0.29
	*Staphylococcus saprophyticus* (MRS)	1	0.29
	*Staphylococcus warneri* (MRS)	1	0.29
Difficult-to-treat (DTT) bacteria	*Enterococcus faecalis*	22	6.38
	*Staphylococcus epidermidis* (MRSE, rifampicin-resistant staphylococcus)	6	1.74
	*Candida albicans*	2	0.58
	*Eschericia coli* (quinolone-resistant Gram-negative bacteria)	2	0.58

**Table 6 antibiotics-13-00524-t006:** Spacer cement and the individual mixture of added anti-infective substances.

	PJI with DTT	PJI with ETT	PJI with MRS
Copal G+C (*n* = 69)	5	63	1
Copal G+V (*n* = 9)	2	6	1
Copal G+C+mixed (*n* = 254)	22	203	29
Copal G+V+mixed (*n* = 7)	1	6	0
Palacos+G+mixed (*n* = 6)	1	3	2
Total	31	281	33

**Table 7 antibiotics-13-00524-t007:** Intravenously administered anti-infective substances or combination of anti-infective substances with number of patients.

Antibiotic One	Antibiotic Two	Antibiotic Three	Antibiotic Four	Number
Vancomycin	Rifampicin			86
Flucloxacillin	Rifampicin			70
Ampicillin/sulbactam	Rifampicin			44
Vancomycin	Imipenem			20
Imipenem	Rifampicin			14
Penicillin G	Rifampicin			13
Cefuroxim	Rifampicin			9
Vancomycin	Fosfomycin			8
Levofloxacin	Rifampicin			7
Imipenem/cilastatin	Rifampicin			6
Daptomycin	Rifampicin			5
Vancomycin	Meropenem			4
Ciprofloxacin	Meropenem			3
Ciprofloxacin	Rifampicin			3
Vancomycin	Imipenem	Rifampicin		3
Vancomycin	Meropenem	Rifampicin		3
Amoxicillin/clavulansäure	Rifampicin			2
Ampicillin/sulbactam	Gentamicin			2
Cefuroxim				2
Flucloxacillin	Piperacillin/tazobactam			2
Meropenem				2
Amoxicillin/clavulansäure	Daptomycin			1
Amoxicillin/clavulansäure	Levofloxacin	Rifampicin		1
Ampicillin	Rifampicin			1
Ampicillin/sulbactam	Fosfomycin			1
Vancomycin	Ampicillin/sulbactam	Fosfomycin		1
Vancomycin	Ampicillin/sulbactam	Gentamicin	Rifampicin	1
Ampicillin/sulbactam	Imipenem			1
Vancomycin	Ampicillin/sulbactam			1
Ampicillin/sulbactam	Metronidazol			1
Ampicillin/sulbactam	Amikacin	Ethambutol	Pyrazinamid	1
Ampicillin/sulbactam	Clindamycin			1
Cephazolin	Clindamycin			1
Ciprofloxacin	Flucloxacillin	Rifampicin		1
Ciprofloxacin	Imipenem			1
Vancomycin	Ciprofloxacin			1
Clindamycin	Rifampicin			1
Vancomycin	Clindamycin			1
Cotrimoxazol	Rifampicin			1
Flucloxacillin	Fosfomycin			1
Flucloxacillin	Imipenem			1
Flucloxacillin	Levofloxacin	Rifampicin		1
Flucloxacillin	Meropenem	Rifampicin		1
Flucloxacillin	Moxifloxacin			1
Vancomycin	Flucloxacillin	Rifampicin		1
Fluconazol	Rifampicin			1
Vancomycin	Fosfomycin	Imipenem		1
Fosfomycin	Meropenem			1
Fosfomycin	Rifampicin			1
Gentamicin	Penicillin G			1
Imipenem	Levofloxacin			1
Levofloxacin	Meropenem			1
Vancomycin	Levofloxacin			1
Levofloxacin	Metronidazol			1
Penicillin G				1
Voriconazol	Rifampicin			1
Vancomycin	Teicoplanin			1
Vancomycin	Piperacillin/tazobactam			1

**Table 8 antibiotics-13-00524-t008:** Administered oral antibiotics, antimycotics, or combination and number.

Antibiotic One	Antibiotic Two	Antibiotic Three	Antibiotic Four	Number
Levofloxacin	Rifampicin			174
Amoxicillin/clavulansäure	Rifampicin			52
Cotrimoxazol	Rifampicin			18
Linezolid	Rifampicin			17
Ciprofloxacin	Rifampicin			13
Amoxicillin/clavulansäure	Levofloxacin			12
Clindamycin	Rifampicin			12
Loracarbef	Rifampicin			6
Ciprofloxacin				4
Clindamyci	Levofloxacin			4
Moxifloxacin	Rifampicin			3
Amoxicillin/clavulansäure	Levofloxacin	Rifampicin		2
Ampicillin	Rifampicin			2
Ampicillin/Sulbactam	Rifampicin			2
Cefuroxim	Rifampicin			2
Ciprofloxacin	Linezolid			2
Clarythromycin	Levofloxacin	Rifampicin		2
Cotrimoxazol				2
Levofloxacin				2
Amikacin	Ethambutol	Pyrazinamid	Rifabutin	1
Amoxicillin/clavulansäure	Metronidazol			1
Amoxicillin/clavulansäure	Clindamycin			1
Cefuroxim				1
Cefuroxim	Clindamycin			1
Ciprofloxacin	Levofloxacin			1
Ciprofloxacin	Linezolid	Rifampicin		1
Cotrimoxazol	Fluconazol			1
Cotrimoxazol	Levofloxacin			1
Cotrimoxazol	Levofloxacin	Rifampicin		1
Fusidinsäure	Rifampicin			1
Levofloxacin	Metronidazol			1
Rifampicin	Voriconazol			1
Rifampicin	Doxycyclin			1

## Data Availability

We do not wish to share our data because of individual privacy, and according to the policy of our hospital the data cannot be shared with others without permission.

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
