# Peer review of "Effect of Multiantibiotic-Loaded Bone Cement on the Treatment of Periprosthetic Joint Infections of Hip and Knee Arthroplasties—A Single-Center Retrospective Study"

_antibiotics, 2024, doi:10.3390/antibiotics13060524_

Round 1

Reviewer 1 Report

Comments and Suggestions for Authors

Overall, I found the article to be well-written and the subject matter highly relevant. However, I have identified several areas that could benefit from revision and refinement

Major points

Introduction part is too short. I suggest to add more data regarding pathogens involved in infections and the historical improvements on these topic to reach the two-stage septic revision method for the treatment of periprosthetic infection with statistic values.

There is a clear overlap between MM and result resctions.

In Figures 1 and 2, the authors discuss the post-treatment survival rate. This could inadvertently suggest a direct correlation between the cause of death and infections, which may not always be the case. It would be beneficial to address this aspect and provide a nuanced discussion on the topic

The three groups - ETT, MRS, and DTT - have been identified as influential factors in antibiotic selection. However, the potential for pathogens to transition from one group to another should be acknowledged and factored into the discussion

In the discussion, the authors address the issue of biofilms. While the use of combined antibiotics is one approach, there are emerging strategies to combat biofilm formation. It would be beneficial to include a discussion on these innovative approaches as future perspectives

Minor points:

Keywords: Consider condensing for brevity and precision

Some sentences are quite long and could be broken down into smaller sentences for clarity. For example, the sentence in discussion starting with “However, Kurd et al [9] showed…” could be split into two sentences for better readability. Also, Despite the sentence:” the larger number of two-stage septic revi- sion surgeries…… the three  pathogen groups”

The reference list includes several older sources. It would enhance the relevance and currency of your article if you could incorporate more recent publications related to your topic

Author Response

Reviewer 1: Thank you for reviewing our paper. The answers are in red.

Overall, I found the article to be well-written and the subject matter highly relevant. However, I have identified several areas that could benefit from revision and refinement

Major points

Introduction part is too short.

We expanded the introduction.

I suggest to add more data regarding pathogens involved in infections

Is done.

and the historical improvements on these topic to reach the two-stage septic revision method for the treatment of periprosthetic infection with statistic values.

Is done.

There is a clear overlap between MM and result resctions.

Is corrected

In Figures 1 and 2, the authors discuss the post-treatment survival rate. This could inadvertently suggest a direct correlation between the cause of death and infections, which may not always be the case. It would be beneficial to address this aspect and provide a nuanced discussion on the topic

Figure 1 and 2 discuss the survival rate after two-step septic revision and not the post-treatment survival rate of the patient. We added information for clarification.

The three groups - ETT, MRS, and DTT - have been identified as influential factors in antibiotic selection. However, the potential for pathogens to transition from one group to another should be acknowledged and factored into the discussion

We analyzed further data to discuss this point.

In the discussion, the authors address the issue of biofilms. While the use of combined antibiotics is one approach, there are emerging strategies to combat biofilm formation. It would be beneficial to include a discussion on these innovative approaches as future perspectives

Is done.

Minor points:

Keywords: Consider condensing for brevity and precision.  Is corrected

Some sentences are quite long and could be broken down into smaller sentences for clarity. For example, the sentence in discussion starting with “However, Kurd et al [9] showed…” could be split into two sentences for better readability. Also, Despite the sentence:” the larger number of two-stage septic revi- sion surgeries…… the three  pathogen groups”

Is done.

The reference list includes several older sources. It would enhance the relevance and currency of your article if you could incorporate more recent publications related to your topic

We added additional sources.

Reviewer 2 Report

Comments and Suggestions for Authors

Blersch et al. conducted a retrospective study to investigate the effect of using multi-antibiotic cement spacers, tailored to pathogen susceptibility, on the survival rate. Although the findings are interesting, they are somewhat predictable, particularly concerning the differences in survival rates across the three microorganism categories. The authors should elaborate on the implications of these findings for research and clinical practice, including future directions and follow-up studies.

Title: In my opinion, the title is not sufficiently informative and does not clearly convey what the authors are specifically investigating. For instance, it's unclear whether the study examines the effect of adding different antibiotics, compares the addition of various antibiotics on outcomes, or analyzes the same antibiotics across different microorganisms. Additionally, it's unclear whether the antibiotics are incorporated into the cement or administered orally.

Methods: The methods section is incomplete. It should include the institution name(s), city, and country where the study was conducted. If this is a single-center study, that information should be explicitly mentioned in the title, abstract, and methods sections. Additionally, the retrospective nature of the study must be stated in the title or abstract.

Table 1: Please include a definition for the asterisk sign in the table footnote.

Comments on the Quality of English Language

I urge the authors to go through the manuscript and correct any grammatical and/or typographical errors.

Author Response

Reviewer 2: Thank you for reviewing our paper. The answers are in red.

Blersch et al. conducted a retrospective study to investigate the effect of using multi-antibiotic cement spacers, tailored to pathogen susceptibility, on the survival rate. Although the findings are interesting, they are somewhat predictable, particularly concerning the differences in survival rates across the three microorganism categories. The authors should elaborate on the implications of these findings for research and clinical practice, including future directions and follow-up studies.

Title: In my opinion, the title is not sufficiently informative and does not clearly convey what the authors are specifically investigating. For instance, it's unclear whether the study examines the effect of adding different antibiotics, compares the addition of various antibiotics on outcomes, or analyzes the same antibiotics across different microorganisms. Additionally, it's unclear whether the antibiotics are incorporated into the cement or administered orally.

We adjusted the title.

Methods: The methods section is incomplete. It should include the institution name(s), city, and country where the study was conducted. If this is a single-center study, that information should be explicitly mentioned in the title, abstract, and methods sections.

Is done.

Additionally, the retrospective nature of the study must be stated in the title or abstract.

Is done.

Table 1: Please include a definition for the asterisk sign in the table footnote.

Is done.

Reviewer 3 Report

Comments and Suggestions for Authors

Dear authors i read the manuscritp with great interest. It is well written and adequate supported.Please consider the following for an updated version

1. Given the fact that PJIs has been proven to be multifactorial please consider adding a paragraph on that topic

2. Please consider adding data on multidrug resistant PJIs

Author Response

Reviewer 3: Thank you for reviewing our paper. The answers are in red.

Dear authors i read the manuscritp with great interest. It is well written and adequate supported.Please consider the following for an updated version

  1. Given the fact that PJIs has been proven to be multifactorial please consider adding a paragraph on that topic

Is done.

  1. Please consider adding data on multidrug resistant PJIs

Is done.

Round 2

Reviewer 2 Report

Comments and Suggestions for Authors

The authors addressed my comments. I endorse the publication in present form.